

# On the spin content of the classical massless Rarita–Schwinger system

**Mauricio Valenzuela⋆ and Jorge Zanelli**

Centro de Estudios Científicos (CECs), Arturo Prat 514, Valdivia, Chile
Facultad de Ingeniería, Arquitectura y Diseño, Universidad San Sebastián, Valdivia, Chile

⋆ mauricio.valenzuela@uss.cl

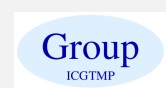

## Abstract

We analyze the Rarita–Schwinger massless theory in the Lagrangian and Hamiltonian approaches. At the Lagrangian level, the standard gamma-trace gauge fixing constraint leaves a spin–$\frac{1}{2}$ and a spin–$\frac{3}{2}$ propagating Poincaré group helicities. At the Hamiltonian level, the result depends on whether the Dirac conjecture is assumed or not. In the affirmative case, a secondary first class constraint is added to the total Hamiltonian and a corresponding gauge fixing condition must be imposed, completely removing the spin–$\frac{1}{2}$ sector. In the opposite case, the spin–$\frac{1}{2}$ field propagates and the Hamilton field equations match the Euler-Lagrange equations.

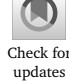

## 1 Introduction

In 1939, Markus Fierz and Wolfgang Pauli discussed the obstacles in the attempt to quantize fields of arbitrary spin $\geq 1$ interacting with photons [1]. Two years later, William Rarita and Julian Schwinger simplified the Fierz-Pauli treatment, writing down a set of field equations describing fermions of arbitrary spin $\geq 3/2$ [2]. The Rarita-Schwinger system (**RS**) describes a field of spin $k + 1/2$ as a tensor-spinor of rank $k$, $\psi^{\alpha}_{\mu_1 \cdots \mu_k}$, symmetric in its tensor indices $\mu_1 \cdots \mu_k$, satisfying a Dirac-like field equation with mass,

$$(\slashed{\partial} + M)\psi_{\mu_1 \cdots \mu_k} = 0, \qquad \gamma^{\mu}\psi_{\mu\mu_2 \cdots \mu_k} = 0. \tag{1}$$

The subsidiary conditions $\partial^{\mu}\psi_{\mu\mu_2 \cdots \mu_k} = 0$ (transverse) and $\psi^{\mu}_{\mu_2 \cdots \mu_k} = 0$ (traceless), appear as consequence of (1) for $M \neq 0$. In the spin–$\frac{3}{2}$ case, of a vector-spinor $\psi^{\alpha}_{\mu}$, Rarita and Schwinger also noted that there is a class of Lagrangians parametrized by the mass ($M$) and a dimensionless coefficient ($A$) that gives rise to the equations (1) (see *e.g.* [3–5]). Then they

chose some $A$ "which permits a relatively simple expression of the equations of motion in the presence of electromagnetic fields".

The description of spin–$\frac{3}{2}$ particles adopted in supergravity, however, traditionally referred to as the Rarita–Schwinger Lagrangian [6,7] (see also [8] and references therein) corresponds to a different choice of $A$ which in the massless limit gives the Lagrangian[1]

$$\mathcal{L} := -\frac{i}{2}\bar{\psi}_\mu \gamma^{\mu\nu\lambda}\partial_\nu \psi_\lambda \,, \tag{2}$$

whose corresponding field equations are (see [9, 10] for the chiral spinor version of these equations)

$$\gamma^{\mu\nu\lambda}\partial_\nu \psi_\lambda = 0 \,. \tag{3}$$

We emphasize that (2) is not equivalent to massless limit of the original RS action since, as shown below, the $\gamma$-trace condition arises as a gauge choice and not as consequence of the field equations.

The action changes by a boundary term and (3) is invariant under the gauge transformation

$$\delta\psi_\mu = \partial_\mu \epsilon \,. \tag{4}$$

Eq. (3) can also be written as

$$\slashed{\partial}\psi_\mu - \partial_\mu \gamma \cdot \psi = 0 \,, \qquad \partial \cdot \psi - \slashed{\partial}\gamma \cdot \psi = 0 \,, \tag{5}$$

and, in the $\gamma^\mu \psi_\mu = 0$ gauge, as

$$\slashed{\partial}\psi_\mu = 0 \,, \qquad \partial^\mu \psi_\mu = 0 \,, \qquad \gamma^\mu \psi_\mu \overset{gf}{=} 0 \,. \tag{6}$$

The third equation corresponds to the gauge choice that fixes the freedom (4), where the symbol $\overset{gf}{=}$ reflects this. In the massive RS system, $\partial^\mu \psi_\mu = 0$ is a consistency condition of the field equations, hence (6) can be obtained from the massless limit of the original RS equations,[2] however (6) can not be obtained by direct variation of the massless action.

The RS field $\psi_\mu$ belongs to the reducible representation $\frac{3}{2} \oplus \frac{1}{2}$ of the Lorentz group and it can be split into its irreducible parts as $\psi_\mu := \rho_\mu + \gamma_\mu \kappa$, where the spin-3/2 part $\rho_\mu$ is gamma-traceless, $\gamma^\mu \rho_\mu = 0$, and $\kappa$ represents the spin-1/2. Then the Euler-Lagrange equations (5) read

$$(D-1)\slashed{\partial}\kappa - \partial^\mu \rho_\mu = 0 \,, \qquad \slashed{\partial}\rho_\mu - \gamma_\mu \slashed{\partial}\kappa - (D-2)\partial_\mu \kappa = 0 \,. \tag{7}$$

Using the gauge freedom (4) it is possible to make $\rho_\mu$, not only gamma-traceless but also divergence-free. Hence, the first equation reduces to the massless Dirac equation for the spin–$\frac{1}{2}$ field $\kappa$, while the second becomes a Dirac equation for the massless spin–$\frac{3}{2}$ with source $\partial_\mu \kappa$. This shows that at least in the gauge $\partial^\mu \rho_\mu = 0$ both spin sectors seem to propagate.

Another way to see that the RS may propagate is the fact that in the vacuum of the spin–$\frac{3}{2}$ field $\rho_\mu = 0$ the RS action produces the Dirac action,

$$\mathcal{L} := i\frac{(D-1)(D-2)}{2}\bar{\kappa}\slashed{\partial}\kappa \,, \tag{8}$$

where $D$ is the number of spacetime dimensions.

---

[1]Here $\gamma_\mu$, $\{\gamma_\mu, \gamma_\nu\} = 2\eta_{\mu\nu}$, are Dirac matrices, $\eta_{\mu\nu} = \texttt{diag}(-1,1,\dots)$ and $\gamma_{\mu\cdots\nu} = \gamma_{[\mu}\cdots\gamma_{\nu]}$ are completely antisymmetric products. We assume the Majorana reality condition $\psi^\dagger = \psi$, $\bar{\psi} = \psi^t C$, were $C^t = -C$.

[2]This is analogous to the transverse condition $\partial^\mu A_\mu = 0$ that is required by consistency of the Proca equations but is only a gauge option in Maxwell's theory.

Though this result might seem straightforward, it should be unexpected if the spin–$\frac{1}{2}$ is pure gauge. One would expect a trivial action principle, as it happens in standard gauge theories. In this paper we look for an explanation to this problem.

By excellence, Hamiltonian analysis is the standard framework for elucidating what are the degrees of freedom of gauge systems. We shall see that either result can be obtained depending on whether or not the validness of the Dirac conjecture—which says that all first class constraints are gauge generators—is assumed. This is a technical observation: if the Dirac conjecture is not assumed, and a gauge fixing condition is impossed only for the primary first class constraint, then the spin–$\frac{1}{2}$ field propagates. Otherwise, the sum of the secondary first class constraint to the extended Hamiltonian introduce a new arbitrary function of time (the corresponding Lagrange multiplier) which, in order to produce a deterministic system of equations, requires an additional gauge fixing condition, which removes the spin–$\frac{1}{2}$ field, in agreement with [8, 11–14].

We shall see exactly in which step of the Dirac algorithm the two branches are generated: the one in which the spin–$\frac{1}{2}$ sector remains and the other where it is removed.

## 2 Space and time splitting

In terms of Poincaré group the vector-spinor $\psi_\mu^\alpha$ can be decomposed in irreducible representation of spins $(1 \oplus 0) \otimes \frac{1}{2} = \frac{3}{2} \oplus \frac{1}{2} \oplus \frac{1}{2}$ [8]. Hence the RS action should be regarded has a spin–$\frac{1}{2}$ and spin–$\frac{3}{2}$ particle systems. The Poincaré spin split can be achieved explicitly in terms of spin-block projectors [8, 15, 16], which involve nonlocal operators. These projectors reveals the gauge invariant components of the RS action and it produces a decoupled system of spin–$\frac{3}{2}$ and spin–$\frac{1}{2}$ governed by Dirac kinetic terms, in agreement with the discussion following (7) (for further details see [17]). The second spin–$\frac{1}{2}$ mode is the pure gauge mode, as it belongs to the kernel of the RS kinetic operator off-shell. The analogous treatment of the Maxwell theory would introduce the transverse (spin-one) and the longitudinal (spin-0) projectors, of which only the transverse mode would propagate.

Non-locality along time directions, however, may be problematic: they might be incompatible with the integration of the field equations under initial conditions on a Cauchy surface. Thus another method is necessary to analyze the problem. Spatial nonlocality, on the other hand, is compatible with the Cauchy data because it leaves the initial surface intact and therefore gauge transformations or field redefinitions involving nonlocal spatial operators such as $\displaystyle{\not\!\nabla}^{-1} := (\gamma^i \partial_i)^{-1}$, should not lead to inconsistencies.

We first split the vector-spinor $\psi_\mu$ into $\psi_0$ and $\psi_i$, which is in turn split into three more pieces: the spatial divergence $\partial^i \psi_i$, the $\gamma$–trace $\gamma^i \psi_i$, and a spatial $\gamma$–traceless and divergenceless vector-spinor $\xi_i$ ($\gamma^i \xi_i = 0 = \partial^i \xi_i$). Thus we have one spin–$\frac{3}{2}$ field $\xi_i$, and three spin–$\frac{1}{2}$ representations of the spatial rotation group, $\psi_0$, $\partial^i \psi_i$ and $\gamma^i \psi_i$. We shall see that the $\gamma$–traceless and divergenceless conditions (6) remove one spin–$\frac{1}{2}$ representations of the rotation group each, whilst the third is a propagating spin–$\frac{1}{2}$ irreducible representation of the Poincaré group.

Consider the decomposition of the identity of vector-spinors in three orthogonal projectors, $\mathbb{1} = P^\perp + P^N + P^L$,

$$(P^N)_{ij} := \frac{1}{D-2} N_i N_j, \qquad (P^L)_{ij} := L_i L_j, \qquad P^\perp = \mathbb{1} - P^N - P^L, \qquad (9)$$

where $N_i := \gamma_i - L_i$ and $L_i := {\not\!\nabla}^{-1} \partial_i$. These are space-like spin-block projectors that decompose

the spatial vector-spinor as

$$\psi_i = \xi_i + N_i \zeta + L_i \lambda, \quad \text{where} \quad \xi_i = P^{\perp\perp}{}_i{}^j \psi_j, \quad \zeta = \frac{1}{D-2} N^i \psi_i, \quad \lambda = L^i \psi_i, \qquad (10)$$

which can be verified with the help of the identities $N_i N^i = D - 2$, $L_i L^i = 1$, $N_i L^i = 0$.

The gamma traceless relation is equivalent to

$$\psi_0 = -\gamma_0 \gamma^i \psi_i = -\gamma_0 ((D-2)\zeta + \lambda), \qquad (11)$$

which, together with the decomposition (10), reduce (6) to

$$\displaystyle{\not{\partial}} \xi_i = 0, \qquad \displaystyle{\not{\partial}} \tilde{\lambda} = 0, \qquad \dot{\zeta} = 0, \qquad \not{\nabla} \zeta = 0, \qquad (12)$$

where $\tilde{\lambda} = \gamma^0 \lambda$. Thus the explicit solution of (6) containing the spin–$\frac{1}{2}$ and spin–$\frac{3}{2}$ sector is,

$$\psi_0 = \gamma^0 \lambda, \qquad \psi_i = \xi_i + \partial_i \not{\nabla}^{-1} \lambda, \qquad (13)$$

given in terms of standard solutions of the Dirac equations (12), restricted by double-transverse condition $\gamma^i \xi_i = \partial^i \xi_i = 0$ [13, 18]. It follows that fields $\xi_i$ and $\lambda$ propagate, whilst $\zeta$ is a constant spinor –and must therefore vanish on-shell–, and $\psi_0$ is not an independent field. Hence, the massless RS equations in the gauge $\gamma^\mu \psi_\mu = 0$ eliminates two spin–$\frac{1}{2}$ modes, $\psi_0$ and $\zeta$, whilst one spin–$\frac{1}{2}$ and one spin–$\frac{3}{2}$ field propagate. Thus, there are no arbitrary functions of time left in the system: the dynamical equations (12) completely determine the evolution of the fields, provided initial data is given on a Cauchy surface.

The number of degrees of freedom in the system is defined by one half the number of functions in the set $\{\xi_i^\alpha(t_0, \vec{x}), \lambda^\alpha(t_0, \vec{x})\}$ necessary to specify the evolution. These functions are: $k \times (D-1) - 2k$ components of $\xi_i^\alpha$, $\alpha = 1, \ldots, k$, and $k = 2^{[D/2]}$ components in $\lambda^\alpha$; in both cases the Dirac equation restricts half of them. In total, we are left with $k(D-2)/2$ degrees of freedom which are equivalent to two massless states of spin–$\frac{1}{2}$ and spin–$\frac{3}{2}$, respectively.

It is relevant now to discuss the picture in Dirac's time honored Hamiltonian analysis [19].

## 3 Hamiltonian analysis

Splitting $\psi_\mu$ as in (10) one obtains, up to boundary terms,

$$\mathcal{L} = -i \bar{\psi}_0 \gamma^{0ij} \partial_i \psi_j + \frac{i}{2} \bar{\psi}_i \gamma^{0ij} \dot{\psi}_j - \frac{i}{2} \bar{\psi}_i \gamma^{ijk} \partial_j \psi_k. \qquad (14)$$

The definition of momenta, $\pi^\mu := \partial \mathcal{L} / \partial \dot{\psi}_\mu$, yields the primary constraints

$$\pi^0 \approx 0, \qquad (15)$$

$$\chi^i := \pi^i - \frac{i}{2} \mathcal{C}^{ij} \psi_j \approx 0, \qquad (16)$$

where $\mathcal{C}^{ij}_{\alpha\beta} := -(C\gamma^{0ij})_{\alpha\beta} = \mathcal{C}^{ji}_{\beta\alpha}$ is invertible, $\mathcal{C}^{ij}_{\alpha\beta}(\mathcal{C}^{-1})^{\beta\kappa}_{jm} := \delta^i_m \delta^\kappa_\alpha$,

$$(\mathcal{C}^{-1})^{\alpha\beta}_{ij} = \left( -\frac{1}{(D-2)} \gamma_i \gamma_j \gamma_0 C^{-1} + \delta_{ij} \gamma_0 C^{-1} \right)^{\alpha\beta}. \qquad (17)$$

The constraint (15) states that $\psi_0$ is a Lagrange multiplier and (16) is a consequence of the first order character of the system. The Hamiltonian, including a linear combination of the primary constraints is

$$H = \int d^{D-1}x \left( i \bar{\psi}_0 \gamma^{0ij} \partial_i \psi_j + \frac{i}{2} \bar{\psi}_i \gamma^{ijk} \partial_j \psi_k + \chi^i_\alpha \mu^\alpha_i + \pi^0_\alpha \mu^\alpha_0 \right), \qquad (18)$$

where $\mu_i$ and $\mu_0$ are arbitrary spinorial Lagrange multipliers. Preservation in time of the primary constraint $\pi^0 \approx 0$ yields a secondary constraint,

$$\dot{\pi}^0 = \{\pi^0, H\} = -\frac{\delta H}{\delta \psi_0} = -iC\gamma^{0ij}\partial_i\psi_j \approx 0 \quad \Leftrightarrow \quad \varphi := -iC^{ij}\partial_i\psi_j \approx 0, \tag{19}$$

which is equivalent to the equation of motion obtained varying (14) with respect to $\psi_0$. Preservation in time of the other primary constraints,

$$\dot{\chi}^i = -(C\gamma^i\gamma_0 C^{-1})\varphi + iC^{ij}\partial_j\psi_0^\beta + iC\partial^i\gamma^j\psi_j - iC\slashed{\nabla}\psi^i + iC^{ij}\mu_j \approx 0, \tag{20}$$

and of the secondary one,

$$\dot{\varphi} = iC^{ij}\partial_i\mu_j \approx 0, \tag{21}$$

yield conditions that determine the Lagrange multipliers $\mu_i$ in terms of the phase space fields, and there are no further constraints. It is easily checked that $\chi^i$ is second class, while $\pi^0$ and the linear combination of constraints

$$\tilde{\varphi} := \varphi + i\partial_i\chi^i \approx 0 \tag{22}$$

are first class. The second class constraint $\chi^i$ will be eventually dropped, leaving $\pi^0 \approx 0$ and $\varphi \approx 0$ as the only remaining first class constraints.

It should be noticed that the secondary constraint $\varphi$ is not purely first class. In particular (21) fixes part of the Lagrange multipliers of the system, whilst $\tilde{\varphi}$ mixes primary and secondary constraints. This will be relevant for the discussion of the Dirac conjecture.

The system can be reduced to the surface of the second class constraints (16) by strongly setting $\chi^i = 0$ and replacing Poisson by Dirac brackets, $\{f, g\}_D := \{f, g\} - \{f, \chi_\alpha^i\}C^{-1}{}_{ij}^{\alpha\beta}\{\chi_\beta^j, g\}$, which in the variables $\psi_0$, $\pi^0$, $\xi$, $\zeta$ and $\lambda$, reads

$$\begin{aligned}
\{f, g\}_D = (-1)^f \int d^{D-1}z \Bigg[ &-i\frac{\delta f}{\delta \xi_i} P_{ij}^{\parallel}\gamma_0 C^{-1}\frac{\delta g}{\delta \xi_j} - i\frac{D-3}{D-2}\frac{\delta f}{\delta \lambda}\gamma_0 C^{-1}\frac{\delta g}{\delta \lambda} \\
&+i\frac{1}{D-2}\left(\frac{\delta f}{\delta \lambda}\gamma_0 C^{-1}\frac{\delta g}{\delta \zeta} + \frac{\delta f}{\delta \zeta}\gamma_0 C^{-1}\frac{\delta g}{\delta \lambda}\right) + \left(\frac{\delta f}{\delta \psi_0^\alpha}\frac{\delta g}{\delta \pi_\alpha^0} + \frac{\delta f}{\delta \pi_\alpha^0}\frac{\delta g}{\delta \psi_0^\alpha}\right) \Bigg],
\end{aligned} \tag{23}$$

and the first class Hamiltonian (18) reduces to

$$H_1 = \int d^{D-1}x \left( i(D-2)\bar{\psi}_0\gamma^0\slashed{\nabla}\zeta - \frac{i(D-2)(D-3)}{2}\bar{\zeta}\slashed{\nabla}\zeta + \frac{i}{2}\bar{\xi}^i\slashed{\nabla}\xi_i + \pi_\alpha^0\mu_0^\alpha \right), \tag{24}$$

where the first class secondary constraint $\varphi \approx 0$ is equivalent to $\slashed{\nabla}\zeta \approx 0$.

The question now is whether one should add this secondary first class constraint to the Hamiltonian as an independent gauge generator. This is equivalent to asking whether the Dirac conjecture (**DC**) holds in this case, namely, whether all secondary first class constraints generate gauge transformations. If the conjecture is valid, the gauge transformations generated by $\varphi$ would require gauge fixing; if that is not the case, $\varphi$ does not generate gauge transformations, it should not be included in the Hamiltonian and no gauge fixing would be required.

One can examine the effect of adding $\varphi$ to the Hamiltonian (24) with a Lagrange multiplier. The time evolution defined by $\dot{f} = \{f, H'\}$, with respect to the *extended Hamiltonian* $H' := H_1 + \tau^\alpha\varphi_\alpha$, is

$$\dot{\xi}_i = -\gamma_0\slashed{\nabla}\xi_i, \qquad \dot{\lambda} = -(D-3)\gamma_0\slashed{\nabla}\zeta + \slashed{\nabla}\psi_0 + \slashed{\nabla}\tau, \tag{25}$$

$$\dot{\psi}_0 = -\mu_0, \qquad \dot{\pi}^0 = 0, \qquad \dot{\zeta} = 0, \qquad \slashed{\nabla}\zeta = 0. \tag{26}$$

The gauge symmetry generated by $\pi^0$ is fixed by specifying $\psi_0$, which can be chosen to implement the standard $\gamma$-traceless condition in (6) as $\psi_0 + \gamma_0 \gamma^i \psi_i \approx 0$. This, together with $\pi^0 \approx 0$, form a pair of second class constraints that can be readily eliminated from the phase space. This gauge choice is accessible since $\pi^0$ generates arbitrary shifts in $\psi_0$ and, in particular, the shift $\delta \psi_0 = -(\psi_0 + \gamma_0 \gamma^i \psi_i)$, renders $\psi'_\mu = \psi_\mu + \delta \psi_\mu$ $\gamma$-traceless. Thus in the phase space spanned by $\xi_i, \zeta, \lambda$, using (11) reduces the system (25) to

$$\dot{\xi}_i = -\gamma_0 \not{\nabla} \xi_i, \qquad \dot{\lambda} = \gamma_0 \not{\nabla} \lambda + \not{\nabla} \tau, \qquad \dot{\zeta} = 0 = \not{\nabla} \zeta. \qquad (27)$$

Assuming the DC as valid, would imply that the evolution of $\lambda$ is indeterminate, from the presence of the arbitrary function of time $\tau$, and therefore an external gauge condition in convolution with the constraint $\varphi \approx 0$ would be necessary. Choosing the gauge condition $\lambda \approx 0$, the stationary condition $\dot{\lambda} \approx 0$ determines the Lagrange multipliers, $\tau = 0$. This removes the spin–$\frac{1}{2}$ sector and the only propagating field is $\xi_i$. In this case, the Hamilton approach, with the Dirac conjecture assumed to be valid, do not match the Euler-Lagrange equations (12).

The framework that matches the Lagrangian approach is the one where the DC is not assumed. Then $\varphi$ would not be regarded as a gauge generator, and it should not be added to $H_1$. This is equivalent to setting $\tau = 0$ in (27), and we reproduce the Euler-Lagrange equations (12) and (7) in the gauge $\partial^\mu \rho_\mu = 0$, whose solution is given by the spin–$\frac{1}{2}$ –spin–$\frac{3}{2}$ system (13).

The two scenarios presented above are consistent. Although in the first case the resulting Hamiltonian evolution is not equivalent to the Lagrangian dynamics, it yields a physical subsystem. There are Lagrangian models whose Hamiltonian formulation leads to secondary first class constraints that do not generate gauge transformations [20, 21, 23]. For those counterexamples to the DC it is still possible to *postulate* the validity of the conjecture without running into inconsistencies. Moreover it has been argued that not adopting the Dirac conjecture might lead to problems in the quantization, which supports the idea that it would be safer to assume the validity of the DC in general [23].

On the other hand, it seems unnecessary to postulate the DC in our case; the resulting system is still consistent and in agreement with the Lagrangian description, and the Dirac bracket (23) does not lead to quantization problems of the sort found in the counterexample of the DC in [23]: the Dirac field can be quantized. In addition, in Chapter 3 of [23] the DC is shown to follow from Dirac's constrained Hamiltonian analysis for dynamical systems in which first and second class constraints do not mix in the process. As noted above (22), this condition does not hold here since the secondary constraint $\{H, \pi^0\} = \varphi = \tilde{\varphi} - i \partial_i \chi^i$ is a linear combination of first class and second class constraints. Furthermore, the constraint $\tilde{\varphi}$ (22) is a mixture of a secondary first class constraint and a second class one. Since it mixes both types, it does not have the form required by the proof of the DC presented in [23]. For a critical discussion on the DC see [24, 25].

If the DC is not valid because some secondary first class constraints do not generate gauge transformations, there is no need to provide a gauge condition for those constraints, and the standard formula for the counting of degrees of freedom [22, 23, 27] generalizes as

$$2 \times \begin{bmatrix} \text{Number of} \\ \text{d.o.f.} \end{bmatrix} = \begin{bmatrix} \text{Dimension of} \\ \text{phase space} \end{bmatrix} - \begin{bmatrix} \text{2nd class} \\ \text{constraints} \end{bmatrix} - 2 \times \begin{bmatrix} \text{1st class} \\ \text{gauge} \\ \text{generators} \end{bmatrix} - \begin{bmatrix} \text{1st class} \\ \text{non-gauge} \\ \text{generators} \end{bmatrix}.$$

Note that the last term on the right hand side could be odd, leading to a paradoxical (possibly inconsistent) quantum scenario. However in systems of spinors, first class constraints have an even number of components and therefore not necessarily inconsistent. For the RS system in 4 dimensions, this counting gives $(16 \times 2 - 12 - 2 \times 4 - 4)/2 = 4$ degrees of freedom, which

correspond to two spin–$\frac{3}{2}$ helicities plus two spin–$\frac{1}{2}$ helicities. In references [8, 11–14], on the other hand, the DC is assumed to be valid, concluding that there are only 2 degrees of freedom, those of a massless spin–$\frac{3}{2}$ field.

# 4 Conclusions

The apparent presence of a propagating spin–$\frac{1}{2}$ mode in the RS system contradicts the expectation that the spin–$\frac{1}{2}$ field is a pure gauge mode. A dynamical spin–$\frac{1}{2}$ mode in RS sounds similar to the claim that there is a propagating spin-0 field in the Maxwell theory. However, in contrast to what happens in gauge theories like Maxwell, Yang-Mills or Chern-Simons when evaluated on a pure gauge configuration like $A_\mu = \Lambda^{-1}\partial_\mu\Lambda$, the RS action neither vanishes nor reduces to a boundary term when evaluated on $\psi_\mu = \gamma_\mu\zeta$ for a generic $\zeta$. This means that configurations $\psi_\mu = \gamma_\mu\zeta$ are not zero-modes of the action, unlike what happens in gauge theories for pure configurations. The reduction $\psi_\mu = \gamma_\mu\zeta$ is precisely what is done in unconventional supersymmetry [28–32], while in supergravity the complementary option is selected by imposing $\gamma^\mu\psi_\mu = 0$ [8].

As for quantization issues, the spin–$\frac{3}{2}$ sector of the massless RS field has been quantized in various approaches [11, 12, 14, 33]. In all of them, both spin–$\frac{1}{2}$ sectors of the Poincaré group decomposition are factored out. Following reference [33]—where it is shown that the massless RS field decomposes in a spin–$\frac{1}{2}$ (pure gauge) sector with 0-norm, and spin–$\frac{1}{2}$ and spin–$\frac{3}{2}$ sectors of positive norm—the massless RS can be quantized à la Gupta-Bleuler factoring out only the zero norm state.

So far we have assumed a flat spacetime, although the generalization to a curved background is straightforward. In the light of these results, it would be interesting to consider supergravity theories without enforcing the validity of the Dirac conjecture, which must contain a spin–$\frac{1}{2}$ excitation along with the gravitino. The spin–$\frac{1}{2}$ sector will inherit the gravity and gauge interactions of the vector spinor, which would generate new supergravity phenomenology.

# Acknowledgements

We warmly thank discussions with L. Andrianopoli, G. Bossard, N. Boulanger, C. Bunster, A. Marrani, R. Matrecano, R. Noris, M. Rausch de Traubenberg, P. Sundell, M. Trigiante for their challenging questions, comments and suggestions.

**Funding information** This work was partially funded by grant FONDECYT 1220862 and USS-VRID project VRID-INTER22/10.

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
