# Peer review of "On the spin content of the classical massless Rarita--Schwinger system"

_SciPost Physics Proceedings, doi:SciPost Phys. Proc. 14, 047 (2023)_

## Round 1 · Referee Report · Jean-Pierre Gazeau (Referee 1) · 2023-1-11

Strengths

1) In their submitted paper, the authors revisit the Dirac conjecture as which a \textit{all first class constraints are gauge generators} by presenting a careful analysis of the massless Rarita-Schwinger equations for 3/2 spin field.
2) They compare Lagrangian and Hamiltonian approaches to the question. Interestingly they prove that both approaches are consistent. Furthermore they match under the condition that the Dirac conjecture is ignored.

Weaknesses

Let me suggest that the authors also consider the appearance of undecomposable representations of the Poincar\'e group in the massless case, and the subsequent Gupta-Bleuler structures in which gauge fields are ignored if one works with cosets in Hilbert space of states (see for instance W. Heidenreich, On solution spaces of massless field equations with arbitrary spin, \textit{J. Math. Phys.} \textbf{27}, 2154 (1986).

Report

This Journal's acceptance criteria are met

Requested changes

A few text corrections should be considered by the authors.
\begin{enumerate}   \item Line above Equation (7): ``is represents'' ?   \item Symbol $D$ is introduced in Equation (7) without giving its meaning.    \item Line 7 below Eq. (8) : ``constraints" and not ``constraint''.    \item In first paragraph of Section 2, ``kernel" and not ``Kernel".   \end{enumerate}

  • validity: high
  • significance: -
  • originality: high
  • clarity: good
  • formatting: perfect
  • grammar: good

Author:  Mauricio Valenzuela  on 2023-01-23  [id 3261]

(in reply to Report 1 by Jean-Pierre Gazeau on 2023-01-11)
Category:
pointer to related literature

Dear Editor:

We thank the referee for the valuable comments and the suggested reference by Heidenreich. We did not know the article of Heidenreich and we think helps with the quantization problem of the system, which we should handle in the next future. Accordingly, we added a paragraph in the Conclusions section, page 7, which reads

"As for quantization issues, the \tralf sector of the massless RS field has been quantized in various approaches \cite{Senjanovic:1977vr,Pilati:1977ht,Fradkin:1977wv,Heidenreich:1986vx}. In all of them, both \half sectors of the Poincar\'e group decomposition are factored out. Following reference \cite{Heidenreich:1986vx}---where it is shown that the massless RS field decomposes in a \half (pure gauge) sector with 0-norm, and \half and \tralf sectors of positive norm---the massless RS can be quantized \`a la Gupta-Bleuler factoring out only the zero norm state."

We hope that with this modification the paper will fulfill the referee requirements.

Sincerely,
the authors.

---

## Round 2 · Author Response

We have added the reference by W. Heidenreich, J. Math. Phys. 27 (1986) 2154-2159.

---

## Round 2 · List of Changes

In section 4, Conclusions, page 7, we added the second paragraph, which reads:

"As for quantization issues, the \tralf sector of the massless RS field has been quantized in various approaches \cite{Senjanovic:1977vr,Pilati:1977ht,Fradkin:1977wv}. In all of them, both \half sectors of the Poincar\'e group decomposition are factored out. Following reference \cite{Heidenreich:1986vx}---where it is shown that the massless RS field decomposes in a \half (pure gauge) sector with 0-norm, and \half and \tralf sectors of positive norm---the massless RS can be quantized \`a la Gupta-Bleuler factoring out only the zero norm state."

which includes the new reference

\bibitem{Heidenreich:1986vx}
W.~Heidenreich,
{\it On solutions spaces of massless field equations with arbitrary spin}, J. Math. Phys. \textbf{27} (1986), 2154-2159
doi:10.1063/1.527037

added to the bibliography list as item [31].

---

## Editorial Decision

published